# Dental Implant Survival and Risk of Medication-Related Osteonecrosis in the Jaws in Patients Undergoing Antiresorptive Therapy: A Systematic Review

**DOI:** 10.3390/ijms26083618

**Published:** 2025-04-11

**Authors:** Armando Crupi, Jacopo Lanzetti, Daniela Todaro, Francesco Pera, Francesco Maria Erovigni

**Affiliations:** Department of Oral Rehabilitation and Maxillofacial Prosthesis, Dental School, University of Turin, Via Nizza 230, 10126 Turin, Italy; armando.crupi@unito.it (A.C.); jacopo.lanzetti@unito.it (J.L.); daniela.todaro@unito.it (D.T.); francesco.pera@unito.it (F.P.)

**Keywords:** dental implants, antiresorptive therapy, ONJ, peri-implant care

## Abstract

The interaction between antiresorptive medication and dental implant procedures remains a subject of concern complicating the decision-making process for clinicians. The aim of the study is to conduct a literature review on the relationship between dental implant placement and the incidence of osteonecrosis of the jaw (ONJ) in patients receiving antiresorptive drugs. The systematic review relied on the PRISMA statement using the PICO tool. The literature search was performed using PubMed, EBSCOhost and Scopus for RCTs, controlled clinical trials and cohort studies. The choice of reference studies was made in a blind process with a 100% agreement rate. For all included studies, quality assessment was performed. The research led to the selection of 608 results. Only five studies were included in the review. Three of the included studies were judged as having a low risk of bias. Dental implants may not be linked to a higher risk of osteonecrosis of the jaw in patients taking low-dose bone-modifying agents. The long-term survival of implants in osteoporotic patients taking oral antiresorptive medication was similar to that in a healthy population and significantly higher than in untreated controls.

## 1. Introduction

There is a steady increase in the prevalence of antiresorptive therapies, particularly bisphosphonates (BPs) and denosumab, due to the significant rise in osteoporosis and other bone-related disorders as the population ages.

These drugs negatively regulate osteoclast activity, decreasing the process of bone resorption. Specifically, bisphosphonates inhibit osteoclast activity, preserving the bone density [1].

Although these treatments are effective in reducing the risk of fractures, they have been associated with medication-related osteonecrosis of the jaws (MRONJ) [2], a serious and often debilitating condition that poses considerable management challenges for dental professionals. The increasing demand for comprehensive dental care underscores the importance of understanding the impact of these therapies on dental procedures, particularly regarding the placement and long-term survival of dental implants.

Dental implants are widely regarded as a successful option for restoring edentulous spaces and improving the quality of life in patients requiring prosthetic rehabilitation.

The process of osseointegration begins with the deposition of proteins, ions and other important biological components, such as polysaccharides and proteoglycans, from the titanium oxide layer [3,4] and continues with the advancement of immune cells, particularly osteoblasts, onto the bone–implant interface. This second phase is characterized by the apposition of new bone [5]. All drugs possessing anti-catabolic properties, such as antiresorptive drugs, could influence the process of osseointegration [6].

Despite the extensive and growing body of literature on MRONJ since its initial description by Marx, the interaction between antiresorptive therapies and dental implant procedures remains a matter of concern, continuing to challenge clinical decision-making in dental practice. Several studies have suggested that patients undergoing antiresorptive therapy may have a relatively low risk of developing osteonecrosis of the jaws (ONJ) following dental implant placement when compared to non-medicated individuals [7]. However, this risk is influenced by multiple factors, including the type of drug, dosage, duration of therapy and the patient’s individual risk profile [8]. Conversely, numerous case reports [9] have documented the occurrence of MRONJ associated with implant-related procedures. (Figure 1).

Consequently, in clinical practice, prosthodontists and oral surgeons are often faced with uncertainty regarding the feasibility of implant-prosthetic surgical interventions in patients undergoing antiresorptive therapy. This knowledge gap has significant negative implications for the quality of dental care provided to this patient population.

The first major consequence is a deterioration in the quality of life for patients who require implant-prosthetic rehabilitation, as they are unable to benefit from the comfort and functionality of fixed or implant-supported prostheses [10].

The second consequence is that some patients choose to delay or altogether avoid initiating antiresorptive therapy following the elimination of oral infectious foci [11], due to their reluctance to forego the possibility of fixed prosthetic rehabilitation. This behavior may result in serious consequences for their overall systemic health.

Therefore, it is essential to assess the long-term survival of dental implants in this specific patient population compared to individuals not receiving such therapies. The current lack of consensus on clinical management underscores the need for a systematic evaluation of the existing evidence.

Clarifying this issue is fundamental to ensuring optimal prosthetic rehabilitation for a growing cohort of patients requiring both antiresorptive therapy and dental implants.

This study aimed to conduct a systematic review of the most recent evidence on this topic to investigate the relationship between dental implant placement and the incidence of ONJ in patients receiving antiresorptive drugs.

The review seeks to synthesize existing research to inform clinical guidelines and aid practitioners in making evidence-based decisions regarding the management of patients requiring dental implants who are undergoing or have undergone treatment with antiresorptive medications. By delineating the potential risks and outcomes associated with dental implants in this population, we aim to enhance patient safety and improve treatment outcomes in dental practice.

## 2. Materials and Methods

The protocol was registered in the international Prospective Register of Systematic Reviews PROSPERO (ID CRD42024552215).

The systematic review relied on the Preferred Reporting Items for Systematic Reviews and Meta-Analyses (PRISMA) statement [12] with the use of the PICO (Population, Intervention, Comparison, Outcome) tool [13] to structure the search question.

Studies were selected for the review following the PICOS criteria as follows:Participants: Patients undergoing therapy with ONJ risk drugs, rehabilitated with dental implants or in need of implant prosthetic rehabilitation;Intervention: Dental implant placement;Comparison: Patients without implant supported prosthesis or not exposed to drugs at risk of MRONJ;Outcome: episode of ONJ and implant survival rate (ISR).

The target question was as follows: “Do patients undergoing or who have undergone antiresorptive drug therapies and dental implants have a higher risk of developing ONJ than healthy patients or patients without implants? How is implant survival in patients undergoing or who have undergone antiresorptive drug therapies?”

### 2.1. Inclusion and Exclusion Criteria

The assessment of the studies was carried out through the analysis of the abstract and subsequently of the full-text. The abstracts of the research results were initially evaluated by two reviewers (A.C. and D.T.). In case of concern, these were included in the analysis of the full-text in order to not exclude potentially relevant articles. The abstract analysis was performed in April 2024.

The following inclusion criteria were applied:At least 10 patients included;Follow-up of at least 3 months for all study projects, except cross-sectional studies;Manuscripts published in English.The following exclusion criteria were applied:

Studies that cannot be classified as randomized controlled trials (RCT), controlled trials (TC), series trials (CSS), cohort studies with retrospective/prospective design, case-control studies or transversal studies.

To be included in the review, full-text articles had to follow the same inclusion criteria, while all studies that did not report drug therapy at risk of ONJ and studies that did not consider patients with dental implants or with a need for implant-supported prosthetic rehabilitations were excluded.

### 2.2. Search Strategy and Data Extraction Process

The literature search was performed on electronic databases, including PubMed, EBSCOhost and Scopus, for RCTs, controlled clinical trials and prospective cohort studies, reporting results with a follow-up of at least 3 months (i.e., short-term follow-up) up to 10 April 2024. To obtain a review of the most recent data in this topic, only articles published between 10 April 2019 and 10 April 2024 and in the English language were considered, and no manual search was conducted. The search used multiple combinations of the following MeSH terms: “Dental Implantation”, “Dental Implants”, “Dental Prosthesis, Implant-Supported”, “Dental Implant Surgery”, “Diphosphonates”, “Bisphosphonate”, “Antiresorptive Drugs”, “Medication related to Osteonecrosis of the Jaw”, “Osteonecrosis of the Jaw”, “Osteonecrosis”, “Jaw Disease”, “Bisphosphonate-Associated Osteonecrosis of the Jaw”, “BRONJ”, “ONJ”, “MRONJ”, “Implant Survival”. The search equations were (“Dental Implantation” OR “Dental Implants” OR “Dental Prosthesis, Implant-Supported” OR “Dental Implant Surgery”) AND (“Diphosphonates” OR “Bisphosphonate” OR “Antiresorptive Drugs” OR “Medication related to Osteonecrosis of the Jaw”) AND (“Osteonecrosis of the Jaw” OR “Osteonecrosis” OR “Jaw Disease” OR “Bisphosphonate-Associated Osteonecrosis of the Jaw” OR “BRONJ” OR “ONJ” OR “MRONJ” OR “Implant Survival”).

Two authors (A.C. and D.T.), experts in oral surgery and in the dental management of patients being treated with antiresorptive drugs, were involved in the literature search. The choice of reference studies was made primarily by filtering for the year of publication, language and type of study and secondly through the evaluation of the abstracts and full-text of the articles, in a blind process.

After evaluating compliance with the inclusion and exclusion criteria, the lists of selected items were compared. As for the screening selection, the concordance between the two evaluators had to be of a Cohen’s Kappa Score between 0.61 and 0.80; any disagreement was resolved by consensus or with a third-party reviewer (F.M.E.). Then, the evaluation of full-text eligibility was carried out by the two reviewers independently, and the process of searching references and citations was carried out to achieve a 100% agreement rate between the two authors. The screening process was conducted using an Excel spreadsheet.

From the included articles, the following data were extracted in duplicate: author’s name, year of publication, study design, aim of the study, sample size, therapy at risk of ONJ, time of administration, follow-up, outcome and conclusions. A standardized form was used to extract data from the included studies.

### 2.3. Quality Assessment of the Included Studies

For all included studies, quality assessment was performed (J.L., A.C., T.D., F.M.E.) according to the Cochrane Reviewers’ Handbook [14], and it consisted of the assessment of six items:Random sequence generation;Allocation concealment;Blinding of participants, personnel and outcome assessors;Handling of incomplete outcome data;Selective outcome reporting;Other sources of bias.

To assess other sources of bias, the CONSORT guidelines for non-pharmacological treatments “http://www.consortstatement.org” (accessed on 15 March 2024) were used with a focus on the following:Information concerning the study design;Calibration;Sample size methods;Statistical methods.

All the items were finally deemed adequate, inadequate or unclear [15].

More specifically, the quality of each cohort study was evaluated according to the Newcastle Ottawa Scale (NOS) for Assessing the Quality of Non-randomized Studies [16] and the representativeness of the exposed subjects, the selection of non-exposed subjects and their comparability and the ascertainment of exposure and outcomes, and the assessments of outcomes were evaluated. Each item has been assessed as adequate, not adequate or not reported.

### 2.4. Data Analysis

Data were extracted by four reviewers (A.C., J.L., D.T., F.M.E.) under the supervision of another additional one (F.P.), using a piloted data-extraction form. Studies without sufficient data to enter the meta-analyses were kept in the systematic review and analyzed qualitatively.

To compare the selected studies, the episodes of ONJ reported by the authors both in the groups of patients with implants who were on or taking anti-resorptive drugs and in the groups of healthy patients with implants were extrapolated, differentiating the type of drug administration or follow-up, where reported by the authors.

## 3. Results

### 3.1. Search

The application of the research equation on PubMed, EBSCOhost and Scopus led to 608 results, and of these, 442 records published over 5 years, 12 records not in English and 11 studies not on humans were discarded. Of the remaining records, only randomized and non-randomized controlled trials (RCTs) and observational studies were considered, excluding 125 articles, and after removing the 4 duplicate records. The remaining 14 records were selected based on the abstract. This selection excluded an additional 7 studies.

Seven articles were fully examined, and two studies [17,18] were excluded at this stage because the reference population does not match the PICO of the review. Finally, data from five studies [19,20,21,22,23] could be extracted. Details of the screening process are reported in Figure 2.

### 3.2. Study Characteristics

Table 1 shows a summary of the included studies. Four out of five studies [19,20,22,23] are retrospective cohort studies, and one [21] is a prospective feasibility study.

Two refs. [19,24] of five studies report that the type of drugs used are oral bisphosphonates for the treatment of osteoporosis in all dosages, forms and formulations. Cheng et al. [20] distinguish between patients who take oral bisphosphonates (alendronate, ibandronate), intravenous bisphosphonates (zoledronate) or subcutaneous monoclonal antibodies (denosumab). Andersen et al. [21] consider patients taking high-dose bisphosphonate (pamidronate, zoledronate, ibandronate), denosumab or their combination, and Penoni et al. [22] observe patients who are under therapy with antiresorptive drugs for osteoporosis.

Three of five studies [19,20,22] do not report the time of antiresorptive drugs administration. The patients included in the study by Andersen et al. [21] took drug therapy for a mean of 25 months (range 3–68 months). The patients observed in the study by Park et al. [23] were on drug therapy at risk of ONJ for at least 1 year.

The follow-up of the retrospective studies [19,20,22,23] ranged from a maximum of 20 years to a minimum of 5 months.

In particular, Park et al. [23] have considered two different follow-up times: the first follow-up period was within 1 year after data collection, while the second follow-up period was from 1 year after data collection until December 2020 (approximately 3 years).

The only prospective study included in the review [21] showed a mean follow-up of 4 months after implant surgery and a mean of 20 days after the abutment operation.

Four of five studies [19,20,22,23] have the episodes of ONJ as a main outcome. Two [20,21] of five studies evaluated the implant survival rate (ISR).

### 3.3. Study Samples

The number of patients included ranged from 27 (49 implants) to 17,916 (from a cohort of 38,230 patients) in therapy with bisphosphonates. One study [22] considered the estimation of oral surgery procedures, including implant surgery.

None of the studies reported on the periodontal or peri-implant status of the included patients and if they were undergoing supportive periodontal/peri-implant care.

### 3.4. ONJ Definitions

Two [19,23] of five studies considered an episode of ONJ when patients with specific diagnostic codes (International Classification of Diseases (ICD-10)) received more than two visits if outpatients or almost two days of recovery for inpatients. Two other studies [20,22] used the definition of the American Association of Oral and Maxillofacial Surgeons (AAOMS) update position: “current or previous treatment with antiresorptive or antiangiogenic agents; exposed bone or bone that can be probed through an intraoral or extraoral fistula in the maxillofacial region that has persisted for longer than eight weeks; and no history of radiation therapy to the jaws or obvious metastatic disease to the jaws”. Andersen et al. [21] evaluated ONJ clinically, reporting a lack of mucosal healing, dehiscence or exposed bone and radiographic signs of ONJ in general.

### 3.5. ISR Considerations

The implant survival rate (ISR) has been evaluated by two studies [20,21] out of five. Cheng et al. [20] followed the implants for a period of up to 20 years and indicated whether the dental implant was still present or not.

Andersen et al. assessed the early implant survival until the 20th day over the implant abutment insertion and prior to the loading of the implant. A clinical evaluation was made by assessing the sound on percussion at the implant (percussion test), non-mobility based on the mobility test and no signs of infection.

### 3.6. Study Outcomes

The results of episodes of ONJ are reported in Table 2.

The study of Andersen et al. [21] was not included in the table because, due to the study design, the control group is missing.

Cheng et al. [20] differentiated the type of drug administration into oral and intravenous administration, and Park et al. [23] evaluated the patient in two separate follow-ups.

Cheng et al. [20] included patients with implants who had never taken bisphosphonates in the control group, while the studies of Ryu et al. [19] and Park et al. [23] had a control group of patients undergoing therapy with bisphosphonates, but without implant rehabilitation.

Only the study by Cheng et al. [20] reported comprehensive data on implant survival. Therefore, a specific meta-analysis on the ONJ or ISR outcome was not performed.

### 3.7. Risk of Bias

The quality assessment of the five included studies was performed according to Newcastle Ottawa Scale (NOS) for Assessing the Quality of Non-randomized Studies [16]. Three [19,20,23] of the five cohort studies included were scored with the maximum score. The results are summarized in Table 3.

For the study design, the representativeness of the exposed subjects and the comparability of exposed and non-exposed groups on the basis of the design or analysis are not reported in the article of Andersen et al. [21].

## 4. Discussion

Five studies were analyzed to assess whether implant treatment increases the risk of ONJ in patients undergoing antiresorptive therapy and to evaluate the cumulative implant survival rate.

Given the aim to focus on the most recent literature, the number of available studies was limited due to strict selection criteria (i.e., studies published within the last five years). Notably, prospective studies were lacking: four out of five were retrospective, and the only prospective study involving high-dose antiresorptive therapy had a very short follow-up period.

Four studies investigated exclusively osteoporotic populations. None of the studies distinguished between postmenopausal and senile osteoporosis, nor did they stratify patients based on the severity of the disease. However, osteoporosis severity can be inferred in Cheng et al. [20], who compared implant survival rates (ISRs) in patients taking oral bisphosphonates versus those on zoledronate or denosumab.

In that study, ISR was significantly higher in treated patients than in untreated ones.

The study populations varied by sex and age. Ryu [19], Cheng [20] and Park [23] analyzed predominantly female cohorts aged over 65 years, although Cheng included a broader age range (45–90 years). The remaining studies [21,22] did not apply sex selection, with age ranges between 40 and 88 years. Penoni [22] did not report patient sex or age, as the study focused solely on invasive oral procedures (IOPs). Cheng et al. [20] found that age did not significantly impact implant survival in patients under antiresorptive therapy or in untreated groups.

Andersen [21], analyzing an oncologic population, reported a balanced distribution between sexes.

In two out of five studies, the ONJ diagnosis was based on ICD-10 codes (M87.1 for drug-induced osteonecrosis; K10.2 for inflammatory jaw conditions). This methodology, linked to retrospective designs, may have introduced selection bias by failing to exclude confounding conditions (e.g., osteomyelitis) and underestimating asymptomatic ONJ cases. Ryu [19] and Park [23] relied solely on healthcare service databases, lacking a clinical confirmation of ONJ. Penoni [22] based the antiresorptive therapy data on patient self-reporting, without documented clinical verification.

The remaining studies adopted the AAOMS ONJ definition, first introduced in 2007 [25].

Numerous systematic reviews have described dental implants as a reliable treatment option, with 10-year survival rates ≥ 95% in healthy individuals [26,27,28,29]. Only one study [20] directly evaluated implant survival in osteoporotic patients, comparing those with and without antiresorptive therapy.

Cheng et al. [20] reported a 94% implant survival rate up to 20 years (CI: 90–96%) in patients receiving oral antiresorptives, significantly higher than the 84% rate (CI: 79–88%) in untreated osteoporotic/osteopenic patients. The healthy control group had an 89% survival rate (CI: 85–92%) (*p* = 0.0005). The same study also found no significant ISR differences between oral and intravenous bisphosphonates or between intravenous bisphosphonates and denosumab.

Siebert et al. [30] reported a 100% implant survival rate in patients receiving annual intravenous zoledronic acid (5 mg). Similarly, Khoury and Hidajat [31] reported only one implant failure (of 71 implants) in patients treated with both oral and intravenous ibandronate. The loaded implant failed after 5 months, and it was successfully replaced.

Watts et al. [32] described one MRONJ case out of 212 patients treated with denosumab, estimating a risk of 0.5%.

These findings suggest that implant surgery is not a significant risk factor for implant failure in osteoporotic patients.

Ryu et al. [19] and Park et al. [23] evaluated ONJ risk in osteoporotic patients undergoing antiresorptive therapy, with and without dental implants. Outcomes were reported as hazard ratios (HRs) and 95% confidence intervals (CIs). Ryu [19] found a lower ONJ risk in patients with implants (HR = 0.51; CI: 0.37–0.71). According to Kaplan–Meier estimates, ONJ developed earlier in patients without implants. Park [23] found no increased ONJ risk in patients with a history of bisphosphonate use who underwent implant surgery (HR = 0.65; CI: 0.46–0.93) or had implants placed (HR = 0.72; CI: 0.57–0.91).

However, these studies had short observation periods (29 and 36 months), while ONJ onset typically occurs within 24 months of intravenous bisphosphonate therapy and after approximately 36 months (average 4.6 years) with oral treatment [33].

Li et al. [34] reported a 23% implant failure rate in 135 patients receiving antiresorptive therapy (445 implants total), with 19.9% of failures attributed to MRONJ. All implants had been placed before the initiation of therapy.

Sher et al. [8] observed a broad range for ONJ onset: 1 to 223 months post-therapy initiation and 0 to 180 months after implant placement.

Even in studies with longer follow-ups, ONJ incidence remained low. Cheng et al. [20] reported a 0.8% ONJ rate, primarily asymptomatic. Implant failures, including those related to ONJ, were managed successfully with new implants (10-year Kaplan–Meier survival: 89%). This suggests that antiresorptive therapy is not the primary cause of implant failure.

Ting et al. [35] did not assess MRONJ incidence in implant patients under antiresorptive therapy. A Japanese study [36] and one from the Mayo Clinic [7] also concluded that implant placement prior to intravenous bisphosphonate use did not increase the MRONJ risk.

Penoni et al. [22] assessed ONJ occurrence based on invasive oral procedures, including implant placements. Over a 9-year follow-up, the MRONJ incidence was 0.02% (1 case in 4603 IOPs), with no ONJ cases linked to implant procedures. In total, MRONJ incidence in patients on oral bisphosphonates undergoing dental procedures was estimated at 0.03% (2 in 6742).

The only prospective study [21] examined high-dose AR therapy in an oncologic population. Of 49 implants placed, none resulted in MRONJ during the short follow-up period (up to 4 months post-surgery). One patient died before abutment placement. Although this study pioneered an implant rehabilitation analysis in high-dose AR patients, its follow-up was insufficient to draw long-term conclusions.

Earlier studies [37,38,39,40] linked an extended therapy duration and comorbidities to an increased ONJ risk in high-dose AR patients.

Only Andersen [21] and Park [23] addressed the AR therapy duration. Mean therapy durations were 25 months (range: 3–68) and ≥1 year, respectively.

Longer bisphosphonate use may increase the ONJ risk due to skeletal accumulation [41,42]. Denosumab, though not retained in bone, has shown a similar or slightly elevated ONJ risk [43,44].

Given the potential severity of ONJ, clinicians must remain informed about current treatment strategies [45,46] and ensure that patients provide signed informed consent [47].

Due to the high age range, comorbidities were assessed to achieve a homogeneous distribution across the cohorts.

Ryu [19] included the analysis of diabetes (18.8%), hypertension (47.9%) and rheumatoid arthritis (1.0%).

The existence of diabetes mellitus or hypertension did not increase the hazard ratios in all the models. Patients with rheumatoid arthritis and a bisphosphonate prescription history instead had a significantly higher incidence of ONJ than those without them. The patients with RA had a 6.80 times higher HR, which was the highest ratio, and those using BPs had a 4.09 times higher HR (*p* < 0.001).

Park [23] identified comorbidities for 1 year before the index date of each subject: hypertension, diabetes mellitus, dyslipidemia, myocardial infarction, stroke and anemia. The influence of comorbidities was not investigated.

Cheng [20] analyzed systemic risk factors, including diabetes, smoking and the use of glucocorticoids, to achieve a homogeneous distribution across the cohorts. The cohorts resulted in homogeneity, except for glucocorticoids. Overall, systemic factors did not significantly affect the survival of the investigated implant in osteoporotic/osteopenic patients, except for diabetes. The presence of diabetes correlated with worsened survival in the general population (neither AR nor osteoporosis). Since data were unavailable regarding the degree of disease control in patients, the observed correlation is likely driven by diabetic patients, whose condition was poorly managed.

Among surgical procedures, Ryu [19] found that tooth extraction, as a risk factor, was associated with a significant increase in the rate of ONJ (0.58% versus. 0.09%, respectively; *p* < 0.001). Tooth extraction showed a higher risk of ONJ more than five times (HR = 5.89).

Also, Park [23] studied the role of tooth extraction and periodontitis in the development of ONJ. The ratio of subjects with a history of periodontitis and tooth extraction in the dental implant group was higher than that in the control group (without implant). For subjects with a history of periodontitis or tooth extraction, the risk for ONJ occurrence in the dental implant group was significantly lower than in the control group in the first year, although it increased over time (HR = 0.75%).

The dental implant is closely related to tooth extraction, which occurs before the surgery in most cases. It can be argued that, since during osseointegration and peri-implant bone homeostasis, bone resorption mediated by osteoclast plays an important role [40,48,49], bisphosphonates may have a negative effect on the surrounding bone due to their inhibition of the osteoclast function [50,51,52].

Conversely, infectious conditions have been suggested as a key factor for the development of ONJ [47,53,54,55,56,57]; implants themselves without inflammation or infection may not increase the ONJ risk, even in patients with a history of bisphosphonates.

Before implant placement, inflammatory or infectious conditions should be managed through tooth extraction or periodontal treatment to obtain intact bone integrity.

Patients who had complications after extraction should be excluded from dental implantation. Only patients with insignificant problems can undergo this surgery.

The control of inflammation should be the first step, and wound-healing-compromising diseases and medications should be considered for osteoporotic patients.

This may suggest that the risk profiles for ONJ occurrence between the selective insertion of dental implants and other dentoalveolar surgery associated with infectious conditions are different [58] and that dental implants may reduce the risk by requiring dental care for implant placement and maintenance.

In addition, the selective use of dental implants may prevent mucosal sores caused by denture pressure, and this may account for the lower risk ratios of dental implants.

This review has several limitations. The inclusion criteria restricted eligible studies to those published within the last five years, limiting the dataset. Five of six studies were retrospective, and three of them had a short follow-up.

Data on total antiresorptive dosage and duration were often missing, and drug types were sometimes reported generically. Also, disease severity of osteoporosis was not reported.

The incidence of ONJ in two studies [19,23] may differ from that recorded in real life since the data were selected from a database based on the demographic and clinical characteristics, so ONJ could not be confirmed by medical records or direct oral examinations, and the severity of ONJ could not be assessed. There may have been confounders not included in the studies that can affect ONJ development. For example, in one study [20], corticosteroid use was significantly different between sub-cohorts, although it is unknown whether the group with higher corticosteroid use had been treated for secondary osteoporosis instead of primary, and no influence on ISR was found.

Only one study [20] assessed ISR for one manufacturer’s implants, without available data to provide adequate comparisons with other implant systems.

Due to heterogeneity in study designs, a meta-analysis was not feasible. Furthermore, no randomized controlled trials (RCTs) were identified, highlighting the need for high-quality prospective research in this area. Prospective studies with detailed information about systemic conditions (type and gravity of osteoporosis or cancer), antiresorptive therapy (molecule, dosage and duration of therapy) and local factors (presence/absence of intraoral inflammatory condition) are needed to investigate, more precisely, the relationship among the different factors.

## 5. Conclusions

The available data suggest that dental implants may not be linked to a higher risk of osteonecrosis of the jaw (ONJ) in patients taking low-dose bone-modifying agents. The prevalence of ONJ related to osteoporosis treatment was found to be very low. Dental implants did not pose a risk, while a significantly higher risk of ONJ was associated with dental extraction in patients receiving antiresorptive therapy. The long-term survival of implants in osteoporotic patients taking oral antiresorptive medication was similar to that of a healthy population and significantly higher than in untreated controls. However, there are still insufficient data on implants in patients taking “high-dose bone modifying agents”, with few cases and short follow-up periods. Despite these encouraging findings, more prospective studies are needed to further investigate this topic. The guidelines for dental practitioners need to be updated periodically based on recent evidence, and they should recommend consultations and collaboration with physicians.

## Figures and Tables

**Figure 1 ijms-26-03618-f001:**
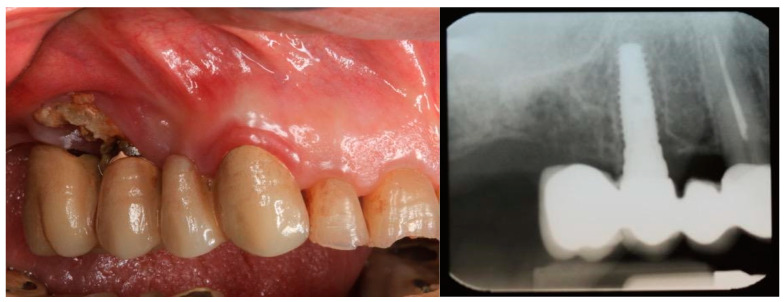
Case of ONJ on 1.5 implant. Clinical image (**left**): a large area of exposed and necrotic bone tissue is visible at dental implant 1.5. Signs of gingival inflammation are also visible. Radiographic image (**right**): the bone structure around the implant 1.5 shows clear signs of low bone density consistent with ONJ.

**Figure 2 ijms-26-03618-f002:**
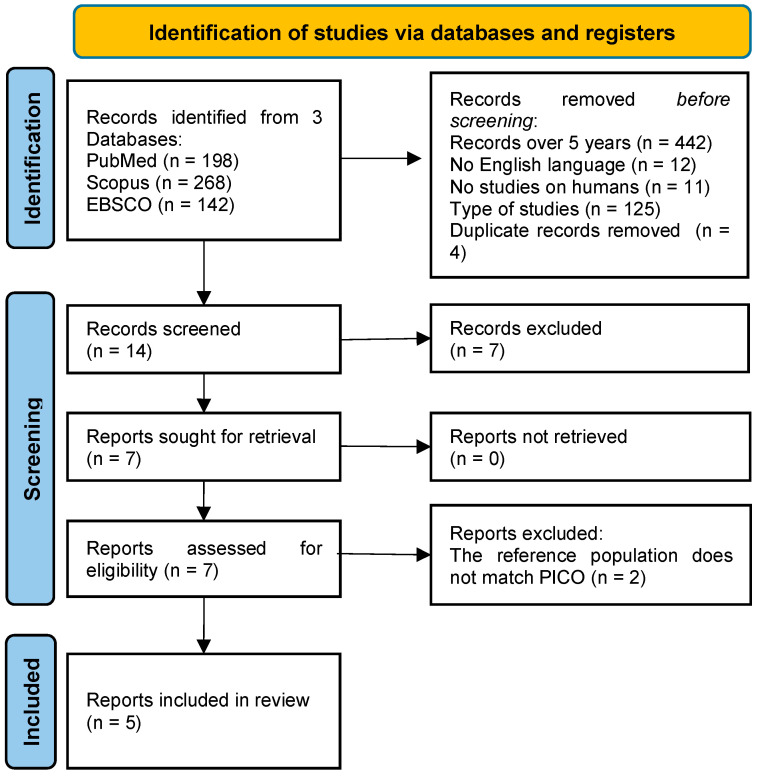
Study flowchart for the search process of the systematic review.

**Table 1 ijms-26-03618-t001:** Characteristics of the included studies.

Authors	Study Design	Aim of the Study	Sample	Therapy at Risk of ONJ	Time of Administration	Follow up	Outcome	Conclusions
Ryu et al., 2021 [19]	Retrospective cohort study	To investigate the association between dental implant therapy and ONJ in osteoporotic patients and the relationships between dental implantation and ONJ	38,230 patients, 17,916 in therapy with bisphosphonates	Bisphosphonates for osteoporosis (pamidronate, alendronate, risedronate, ibandronate and zoledronic acid, all dosages, forms and formulations)	Not reported	2.5 years	ONJ	Dental implantation was not a risk. A significantly higher risk of ONJ was associated with dental extraction in patients under BP therapy
Cheng et al., 2022 [20]	Retrospective cohort study	To investigate the effects of antiresorptive treatment on the survival of plateau-root form dental implants	631 patients, 1472 implants	Oral bisphosphonates (alendronate, ibandronato), intravenous bisphosphonates (zoledronate) or subcutaneous monoclonal antibodies (denosumab)	Not reported	Up to 20 years	ISR *, ONJ	Long-term plateau-root form implant survival in osteoporotic patients taking oral antiresorptives was similar to a healthy population and significantly higher than the untreated controls
Andersen et al., 2023 [21]	Prospective feasibility study	To evaluate the early outcome of dental implant placement in patients with cancer on high-dose antiresorptive medication	27 patients, 49 implants	High-dose bisphosphonate (pamidronate, zoledronate, ibandronate), denosumab or their combination	Mean of 25 months (range 3–68 months)	Mean of 4 months after implant surgery and mean of 20 days after abutment operation	ISR *, ONJ	It is feasible to insert dental implants and perform an abutment surgery in patients with cancer on high-dose antiresorptive medication, without the development of ONJ
Penoni et al., 2023 [22]	Retrospective cohort study	To report the experience of medication-related osteonecrosis of the jaws in osteoporotic patients for nine years and their associated initiating factors and to investigate the possible association of the number of teeth and systemic risk factors	173 procedures (dental implant) estimated	Antiresorptive drugs for osteoporosis	Not reported	9 years	ONJ	The prevalence of ONJ associated with osteoporosis treatment was very low. There are no cases of ONJ caused by implant placement
Park et al., 2024 [23]	Retrospective cohort study	To investigate the risk of implant surgery and implant presence for ONJ occurrence in osteoporotic patients	332,728 patients	Bisphosphonates (pamidronate, alendronato, risedronato, zoledronate, ibandronate) all dosage, forms and formulations	1 year or less	3 years	ONJ	The results may suggest that dental implants are not associated with an increased risk of ONJ; rather, they showed a lower risk.

* ISR: implant survival rate; MBL: marginal bone loss, ONJ: episode of osteonecrosis of the jaw.

**Table 2 ijms-26-03618-t002:** Data of the studies included.

Authors (Years)	ONJ Test Group	Total Sample Test Group (Osteoporotic)	ONJ Control Group	Total Sample Control Group (Osteoporotic)	Medium Follow-up	Statistical Analysis
Cheng et al., 2022 [20]	2	124 (BF+, Imp+)	0	199 (BF−, Imp+)	up to 20 years	The survival of implants placed in patients taking oral antiresorptive medications, 94% (CI: 90–96%), was significantly better than those in the osteoporosis/osteopenia control (84%; CI: 79–88%) (*p* value = 0.0005) in the general population (89%, CI: 85–92%) Univariate analysis showed that improved implant survival was correlated with oral antiresorptive treatment (*z*-value = −2.99, *p* value = 0.03).
Ryu et al., 2021 [19]	41	9738 (BF+, Imp+)	11	12,712 (BF+, Imp−)	Up to 2.5 years	HR = 0.51
Park et al., 2024 [23]	56	29,056 (BF+, Imp+)	105	87,322 (BF+, Imp−)	Up to 1 year	HR = 0.65

Note: ONJ: the episode of osteonecrosis in the jaws; BF+: patients receiving bisphosphonate therapy; Imp+: patients with a dental implant; BF−: patients not receiving bisphosphonate therapy; Imp−: patients without a dental implant; HR < 1 protective effect of the implants on ONJ occurrence, HR > 1, higher risk of ONJ occurrence if an implant is present.

**Table 3 ijms-26-03618-t003:** Details of the risk of bias for cohort studies according to Newcastle Ottawa Scale (NOS) for Assessing the Quality of Non-randomized Studies scale [16].

Authors (Years)	Study Design	Representativeness of the Exposed Subjects	Selection of Non-Exposed Subjects	Ascertainment of Exposure	Ascertainment of Outcome	Comparability of Exposed and Non-Exposed Groups on the Basis of the Design or Analysis	Assessment of Outcome
Ryu et al., 2021 [19]	Retrospective cohort study	Adequate	Adequate	Adequate	Adequate	Adequate	Adequate
Cheng et al., 2022 [20]	Retrospective cohort study	Adequate	Adequate	Adequate	Adequate	Adequate	Adequate
Andersen et al., 2023 [21]	Prospective feasibility study	Adequate	Not required	Adequate	Adequate	Not required	Adequate
Penoni et al., 2023 [22]	Retrospective cohort study	Not adequate	Not adequate	Not adequate	Not adequate	Not adequate	Not adequate
Park et al., 2024 [23]	Retrospective cohort study	Adequate	Adequate	Adequate	Adequate	Adequate	Adequate

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
