# Peer review of "Dental Implant Survival and Risk of Medication-Related Osteonecrosis in the Jaws in Patients Undergoing Antiresorptive Therapy: A Systematic Review"

_ijms, 2025, doi:10.3390/ijms26083618_

Round 1
Reviewer 1 Report
Comments and Suggestions for Authors
This systematic review explores the relationship between dental implant survival rates and the risk of medication-related osteonecrosis of the jaw (MRONJ) in patients undergoing antiresorptive therapy (AR). Following PRISMA guidelines, a literature search was conducted in PubMed, EBSCOhost, and Scopus, identifying six eligible studies, including five retrospective cohort studies and one prospective study. The selection of topics for this article is of clinical value and may provide an important reference for appropriate clinical decisions, but there are still some areas for improvement。
- The heterogeneity of the included studies was high (e.g., different ONJ definitions, drug types, and follow-up times), while the studies did not adequately consider the effects of factors such as age, gender, osteoporosis severity, and duration of anti-bone resorption drug use on the results, and it is recommended that the explanatory power of the results be enhanced by subgroup analyses or sensitivity analyses.
- The long-term effects of high-dose AR medications remain unclear,prospective cohort studies of long-term intravenous drug use in cancer patients need to be supplemented to clarify the quantitative dose-course-risk relationship.
- Some of the retrospective studies suffered from selection bias (e.g., not controlling for confounders), and their impact on the conclusions should be analyzed in more depth.
- It is recommended to write the materials and methods before the results so that the logic of the writing is clearer.
- The Discussion part must be written. The paragraphs are so scattered that there is no logical connection between them. Just repeat the results of studies by other researchers.
The English could be improved to more clearly express the research.
Author Response
- The heterogeneity of the included studies was high (e.g., different ONJ definitions, drug types, and follow-up times), while the studies did not adequately consider the effects of factors such as age, gender, osteoporosis severity, and duration of anti-bone resorption drug use on the results, and it is recommended that the explanatory power of the results be enhanced by subgroup analyses or sensitivity analyses.
Thanks for the comment. I analyzed this aspect in the discussion
- The long-term effects of high-dose AR medications remain unclear prospective cohort studies of long-term intravenous drug use in cancer patients need to be supplemented to clarify the quantitative dose-course-risk relationship.
Thanks for the comment. I added this important comment among the limitations of the study.
- Some of the retrospective studies suffered from selection bias (e.g., not controlling for confounders), and their impact on the conclusions should be analyzed in more depth.
Thanks for the comment. I analyse the impact of the selection bias in the discussion.
- It is recommended to write the materials and methods before the results so that the logic of the writing is clearer.
Thanks for the comment. I put the materials and methods before the results as requested.
- The Discussion part must be written. The paragraphs are so scattered that there is no logical connection between them. Just repeat the results of studies by other researchers. Thanks for the comment. I rewrote the thread following the reviewers' advice.
Reviewer 2 Report
Comments and Suggestions for Authors
This research is under the scope of this journal; the topic is relevant for readers, and this research deals with potentially significant knowledge to the field.
Introduction:
The introduction provides a well-structured overview of the relevance of antiresorptive therapy in implantology. However, it would benefit from an expanded discussion on the mechanisms by which antiresorptive agents influence bone remodeling and wound healing post-implantation.
Please consider recent findings from PMID: 32486396 and PMID:27475683 could add depth to the discussion on how bisphosphonates and denosumab differentially affect osseointegration and the risk of MRONJ. please add this reference:
Language needs improvement, especially in the Introduction and the Discussion. One of the major concerns here is the English. The authors should have the manuscript looked at for language and sentence composition. There are a lot of sentences in the manuscript that do not make sense because of the English.
The study presents valuable findings, particularly regarding the implant survival rate (ISR) in patients receiving oral bisphosphonates versus those on intravenous agents. However, the manuscript should clarify why some studies reported a protective effect of implants against ONJ (as seen in Park et al., 2024), while others found no significant difference. A comparative analysis of methodologies across studies, including the definition of ONJ used in each case, would enhance clarity. Additionally, the figures should be formatted consistently to improve data visualization.
Discussion: please discuss the heterogeneity of included studies
Figure legends: Bad descriptions
There are many mistakes in the references section and in the text
Comments on the Quality of English LanguageLanguage needs improvement, especially in the Introduction and the Discussion. One of the major concerns here is the English. The authors should have the manuscript looked at for language and sentence composition. There are a lot of sentences in the manuscript that do not make sense because of the English.
Author Response
Introduction:
- The introduction provides a well-structured overview of the relevance of antiresorptive therapy in implantology. However, it would benefit from an expanded discussion on the mechanisms by which antiresorptive agents influence bone remodeling and wound healing post-implantation. Thanks for the comment. I added what was requested in the introduction.
- Please consider recent findings from PMID: 32486396 and PMID:27475683 could add depth to the discussion on how bisphosphonates and denosumab differentially affect osseointegration and the risk of MRONJ. please add this reference:
Thanks for the comment. I added this important references in the manuscript.
- Language needs improvement, especially in the Introduction and the Discussion. One of the major concerns here is the English. The authors should have the manuscript looked at for language and sentence composition. There are a lot of sentences in the manuscript that do not make sense because of the English.
Thanks for the comment. I corrected the text grammatically.
- The study presents valuable findings, particularly regarding the implant survival rate (ISR) in patients receiving oral bisphosphonates versus those on intravenous agents. However, the manuscript should clarify why some studies reported a protective effect of implants against ONJ (as seen in Park et al., 2024), while others found no significant difference. A comparative analysis of methodologies across studies, including the definition of ONJ used in each case, would enhance clarity. Additionally, the figures should be formatted consistently to improve data visualization.
Thanks for the comment. I analyzed this aspect in the discussion
Discussion: please discuss the heterogeneity of included studies Thanks for the comment. I analyzed this aspect in the discussion
Figure legends: Bad descriptions
Thanks for the comment. I rewrote the figure legends
There are many mistakes in the references section and in the text Thanks for the comment. I have reviewed all the references.
Reviewer 3 Report
Comments and Suggestions for Authors
The manuscript is entitled " Dental implant survival and risk of medication-related osteonecrosis in the jaws in patients undergoing antiresorptive therapy: a systematic review" The study addresses an important clinical issue, dental implant survival in patients undergoing antiresorptive therapy.
Comments:
- The rationale behind this study is well-established.
- Citations [4,5] are duplicated. Please correct and ensure reference accuracy.
- The exclusion criteria should not be the opposite of the inclusion criteria, such as "Studies with less than 10 patients included"
- What was the reason to choose between April 10th 2019 and April 10th 2024?
- The PRISMA guidelines and PICO framework are appropriately used to structure the review.
- The manuscript refers to a study by Andersen et al. (2023) as both a "prospective feasibility study" and a "prospective cohort study." The authors should ensure consistency.
- The manuscript does not provide clear guidance for clinicians regarding implant placement in patients undergoing antiresorptive therapy. A section on "Clinical Implications" or "Recommendations for Practice" could improve its applicability.
- While the discussion mentions some limitations, the section could be expanded to include: Potential publication bias, Differences in antiresorptive drug dosage and duration across studies.
Author Response
1.The rationale behind this study is well-established.
Thanks for the comment.
2.Citations [4,5] are duplicated. Please correct and ensure reference accuracy.
Thanks for the comment. I have reviewed all the references.
3.The exclusion criteria should not be the opposite of the inclusion criteria, such as "Studies with less than 10 patients included"
Thanks for the comment. I have modified the exclusion criteria in the text.
4.What was the reason to choose between April 10th 2019 and April 10th 2024? Thanks for the comment. I have specified our selection choice in the text.
5.The PRISMA guidelines and PICO framework are appropriately used to structure the review.
Thanks for the comment.
6.The manuscript refers to a study by Andersen et al. (2023) as both a "prospective feasibility study" and a "prospective cohort study." The authors should ensure consistency.
Thanks for the comment. I changed the type of study in the text.
6.The manuscript does not provide clear guidance for clinicians regarding implant placement in patients undergoing antiresorptive therapy. A section on "Clinical Implications" or "Recommendations for Practice" could improve its applicability.
Thanks for the comment. I added some clinical advice to the discussion.
7. While the discussion mentions some limitations, the section could be expanded to include: Potential publication bias, Differences in antiresorptive drug dosage and duration across studies.
Thanks for the comment. I added this limitation in the discussion.
Round 2
Reviewer 1 Report
Comments and Suggestions for Authors
The manuscript has been much improved. However, the discussion section appears somewhat superficial. Although the authors mentioned the inability to perform a meta-analysis due to heterogeneity, a more detailed explanation, including potential subgroup analyses, should be provided to clearly outline the limitations and future research directions.
Author Response
Comment 1:
The manuscript has been much improved. However, the discussion section appears somewhat superficial.
Although the authors mentioned the inability to perform a meta-analysis due to heterogeneity, a more
detailed explanation, including potential subgroup analyses, should be provided to clearly outline the
limitations and future research directions.
Response_1:
We thank the reviewer for his comments. We improved the
discussion as suggested, trying to deepen the issue related to the heterogeneity of the studies.
